

# Gradual, temperature-induced change of secondary sexual characteristics in *Trichogramma pretiosum* infected with parthenogenesis-inducing *Wolbachia*

Su-fang Ning*, Jin-cheng Zhou*, Quan-quan Liu, Qian Zhao and Hui Dong

College of Plant Protection, Shenyang Agricultural University, Shenyang, Liaoning, China
* These authors contributed equally to this work.

## ABSTRACT

Intersex is an intermediate stage of sexual differentiation in insects. Determining intersex morphology and the cause of its production will aid in the understanding of the mechanism of sexual differentiation in insects. In this paper, *Wolbachia*-infected *Trichogramma pretiosum* (*T. preW*$^+$) that shows thelytokous parthenogenesis were used as subjects. In order to determine the causes of the *T. preW*$^+$ intersex and the influence of parental generation temperature on gradual changes in secondary masculinization in intersex offspring, we examined the occurrence of intersex offspring ($F_1$ and $F_2$ generation) after the parental generations were treated with high temperature (27, 29, 31, and 33 °C) and described the external morphology of the intersexes. The results showed that the *T. preW*$^+$ parental generation temperature is positively correlated with the probability of intersex offspring. The probability of $F_1$ intersex is significantly higher than that of $F_2$ intersex in different high temperature. The degree of secondary masculinization in *T. preW*$^+$ intersexes increases as parental generation temperature increases. In addition, our study first identified 11 intersex types in *T. preW*$^+$ and found that the primary and secondary sexual characteristics showed a regular distribution. We also found that the D type and H type of intersex have the highest frequency of appearance. The external genitalia of most intersexes were female, and only three intersex types have male external genitalia. Conclusions were ultimately obtained: *Wolbachia* is a direct factor that causes the occurrence of intersexes, while high temperature is an indirect factor that determines the external morphology of intersexes. The effects of high temperature on *T. preW*$^+$ intersexes is passed through the parental generation to offspring, and this maternal effect weakens as the number of generations increases. In *T. preW*$^+$ intersex individuals, most exhibit female primary sexual characteristics, and secondary sexual characteristics exhibit signs of masculinization.

Corresponding author
Hui Dong, biocontrol@163.com

## INTRODUCTION

The difference between the sexes of the same species is called sexual dimorphism in nature. Generally, two types of sexual characteristics are used to describe sexual dimorphism. Primary sexual characteristics include male and female reproductive organs that are directly related to reproduction (such as external genitalia and internal genitalia), whereas the secondary sexual characteristics include body size, external morphology, antenna, etc., and are used to describe characteristics that aid in reproduction (*Schärer, Rowe & Arnqvist, 2012*; *Tulgetske & Stouthamer, 2012*; *Bear & Monteiro, 2013*). Usually, sexually aberrant individuals with conflicting primary and secondary sexual characteristics are known as intersexes, gynandromorphs, and sexual mosaics (*Bowen & Stern, 1966*; *Pintureauand & Bolland, 2001*; *Pintureau et al., 2002*; *Tulgetske & Stouthamer, 2012*). Of these terms, intersexes refer to individuals with uniform genetic constitution, but some tissues show sexual phenotypes that conflict with the associated genotype (*Goldschmidt, 1949*). However, this concept is frequently confused or wrongly used with gynandromorphs (*Tulgetske & Stouthamer, 2012*). In actual fact, the latter refers to individuals that are composed of genetically distinct tissues that correspond to the sexual phenotypes observed (*Brust, 1966*). An example is the genus *Trichogramma* in the order Hymenoptera, sexually aberrant individuals were often described as gynandromorphs and intersexes. However, *Tulgetske & Stouthamer (2012)* showed that these individuals are intersexes and not gynandromorphs. The intersexes can be falls into two categories: female intersex and male intersex in Hymenoptera (*Whiting, Greb & Speicher, 1934*; *Tulgetske & Stouthamer, 2012*). Female intersexes begin their development as females but organs developed later are masculinized, and male intersexes begin their development as males but organs developed later are feminized (*Whiting, Greb & Speicher, 1934*). As intersex is an intermediate stage of sexual differentiation in insects (*Bowen & Stern, 1966*), in-depth research on these rare sexually aberrant individuals is necessary and crucial. Therefore, determining intersex morphology and the cause of its production will aid in the understanding of the mechanism of sexual differentiation in insects.

Many studies showed that intersex formation may be affected by single or multiple regulatory factors that participate in sex determination. These regulatory factors include mutations (*Cline, 1984*; *Schutt & Nöthiger, 2000*), chromosomal aberrations (*Seiler, Puchta & Brunold, 1958*; *Beukeboom & Kamping, 2006*), and epigenetic factors (e.g., environment, sex hormones, and symbionts) (*Rigaud & Juchault, 1998*; *Pereira et al., 2010*; *Tulgetske & Stouthamer, 2012*; *Martin & Scholtz, 2012*). In *Trichogramma*, intersex formation is associated with *Wolbachia* regulation. The intracellular symbiotic bacteria, *Wolbachia*, is a type of sex regulatory element that is ubiquitous in arthropods. *Wolbachia* can induce insects to undergo parthenogenesis (PI), cytoplasmic incompatibility, male-killing, and feminization (*Werren & Windsor, 2000*; *Almeida & Stouthamer, 2018*). The sex determination mechanism for *Trichogramma* is haplodiploidy, where fertilized eggs develop into diploid females and unfertilized eggs develop into haploid males (*Perlman, Kelly & Hunter, 2008*). However, PI *Wolbachia* can induce parthenogenesis in *Trichogramma* by causing unfertilized eggs to undergo diploidization and feminization.

Thus, females could produce nearly 100% female offsprings even without mating (*Ma et al., 2015*). According to our knowledge of *Trichogramma*, sexually aberrant individuals are only present in parthenogenetic species. (*Tulgetske & Michelle, 2010*). The *Wolbachia*-mediated phenotype are closely related with *Wolbahcia* titers. Many studies revealed *Wolbachia* were sensitive to multiple factors (e.g., temperature and antibiotic treatment) (*Tulgetske & Michelle, 2010*; *Tulgetske & Stouthamer, 2012*; *Almeida & Stouthamer, 2018*). The reduction of *Wolbachia* titers was often reported in high temperature (*Zchori-Fein, Gottlieb & Coll, 2000*; *Ma & Schwander, 2017*). Almost no intersexes appear under normal temperature. However, the number of intersexes significantly increases after high temperature or antibiotic treatment (*Pintureauand & Bolland, 2001*; *Tulgetske & Michelle, 2010*). The specific reason for this may be that *Wolbachia* regulation of *Trichogramma* wasps dominate at normal temperatures but high temperature and antibiotics eliminated the dominance of *Wolbachia*, thereby decreasing *Wolbachia* titers in infected parthenogenetic insects (*Schilthuizen & Stouthamer, 1997*; *Kent et al., 2011*; *Fukui et al., 2015*). This disrupts or inhibits *Wolbachia*-induced feminization (*Rigaud & Juchault, 1998*), ultimately resulting in a higher proportion of intersex offspring (*Pintureauand & Bolland, 2001*; *Pintureau, Chapelle & Delobel, 1999*). Intersexes were reported at frequencies of parental generation after continuous high temperature treatment (*Pintureau, Chapelle & Delobel, 1999*; *Pintureauand & Bolland, 2001*). However, whether parental generation treatment will affect intersex offspring and whether parental generation effects can be transmitted to the next generation are still unknown. In addition, there is a lack of objective and accurate estimation of gradual changes in secondary sexual characteristics as influenced by temperature in intersexes. In view of the aforementioned phenomena, we carried out the following study.

*Trichogramma* is an extremely effective natural enemy of egg parasites and has an indispensable role in biological pest control (*Sigsgaard et al., 2017*). In this study, *Wolbachia*-infected *Trichogramma pretiosum* ($T. preW^+$) populations were used as subjects to study the effects of different feeding temperature in the parental generation (27, 29, 31, 33 °C) on the probability of intersexes in the $F_1$ and $F_2$ generation. The levels of masculinization, which were determined based on previous descriptions (*Tulgetske & Stouthamer, 2012*), were introduced to reveal the effector pattern of high temperature on gradual changes in secondary sexual characteristics in $T. preW^+$. Our study systematically described and identified the external morphological characteristics of intersexes. The results of this study may aid in the understanding of sex determination and sexual differentiation in $T. preW^+$.

## MATERIALS AND METHODS

### Insect breeding

*Trichogramma pretiosum* Riley infected with *Wolbachia* were provided by the Pest Biological Control Laboratory of Shenyang agricultural university. And the uninfected with *Wolbachia* of *Trichogramma pretiosum* Riley (*T. preW*) were obtained by feeding starved 1-day-old wasps a 2 mg/ml solution of the antibiotic, rifampicin, mixed with honey. The population has been maintained in our laboratory for a long time over 20 generations

(ca. 200 days) after collection from the field. The presence of *Wolbachia* in the females was detected by using forward primer 5′-TGGTCCAATAAGTGATGAAGAAAC-3′ and reverse primer 5′-AAAAATTAAACGCTACTCCA-3′ for the *wsp* gene of *T. pretiosum* strains. The wasps were reared on fresh eggs of *Corcyra cephalonica* Stainton (<24 h) that were scattered on paper cards coated with Arabic gum and irradiated with ultraviolet lamp (30W, TUV30W; Philips, Amsterdam, Netherlands) for 30 min. A group of 30 females were introduced in a finger tubes ($1.8 \times 5$ cm) containing a host egg card with ca. 600 *C. cephalonica* eggs for 2 h. Thereafter, the females were removed and the host eggs were reared for ca. 10 days until the emergence of wasps. The *T. preW*[+] have been cultured in laboratory conditions ($25 \pm 1$ °C, $70\% \pm 5\%$ relative humidity with a 16:8 h light:dark photoperiod) in growth chamber (MLR-352H-PC; Panasonic, Osaka, Japan) before use.

## Maternal temperature treatment

Four elevated temperatures: 27, 29, 31, 33 °C, were used in this study. And three repetitions were performed for each temperature treatment. In addition, no intersex were found at 25 °C in *T. preW*, so they are not described in this paper. Take *T. preW* as the control group. *T. preW*[+] is the experimental group.

We firstly selected 25 newly emerged females for each temperature and evenly separated them into five finger tubes ($1.8 \times 5$ cm) with surplus *C. cephalonica* eggs glued on paper cards. These wasps were allowed to lay eggs within 2 h at room temperature. After that, the finger tubes were placed at 27, 29, 31, and 33 °C in the growth chamber (MLR-352H-PC; Panasonic, Osaka, Japan). The subsequently emerged wasps ($F_0$ generation) were maintained under laboratory conditions and supplied with superfluous *C. cephalonica* eggs and 50% honey-water. After 3 days, the wasps were gently removed. Then the emerged $F_1$ generation wasps were allowed to reproduce to get $F_2$ generation. The 1-day-old wasps of $F_1$ or $F_2$ were killed under $-80$ °C conditions for observation. The intersexes were discriminated under the stereomicroscope lens (SMZ-161; Motic, Xiamen, China). The external characteristics of intersex individuals from $F_1$ and $F_2$ generations were observed and described, and the intersex number in each generation was counted. The intersex individuals were determined based on external characteristics with Tulgetske's method (*Tulgetske & Stouthamer, 2012*). Based on the characteristics of antennae and external genitals, we were able to classify the intersexes into 11 morphological categories (from level 1 to level 11 which was labeled by A–K, respectively) with increasing masculinization (Fig. 1). The number of intersex in each level was counted as well.

## Sample preparation and photograph

In order to obtain the pictures of all masculinization levels of intersex and make the size of the sample remains unchanged for a long time, the intersex samples were fixed 2 h in Carnoy's solution (methanol:glacial acetic acid = 3:1). A step by step dehydration was performed with different gradient ethanol, 10–20 min each. Subsequently, the solution in different proportions of ethanol and glycerol was treated step by step for only one time

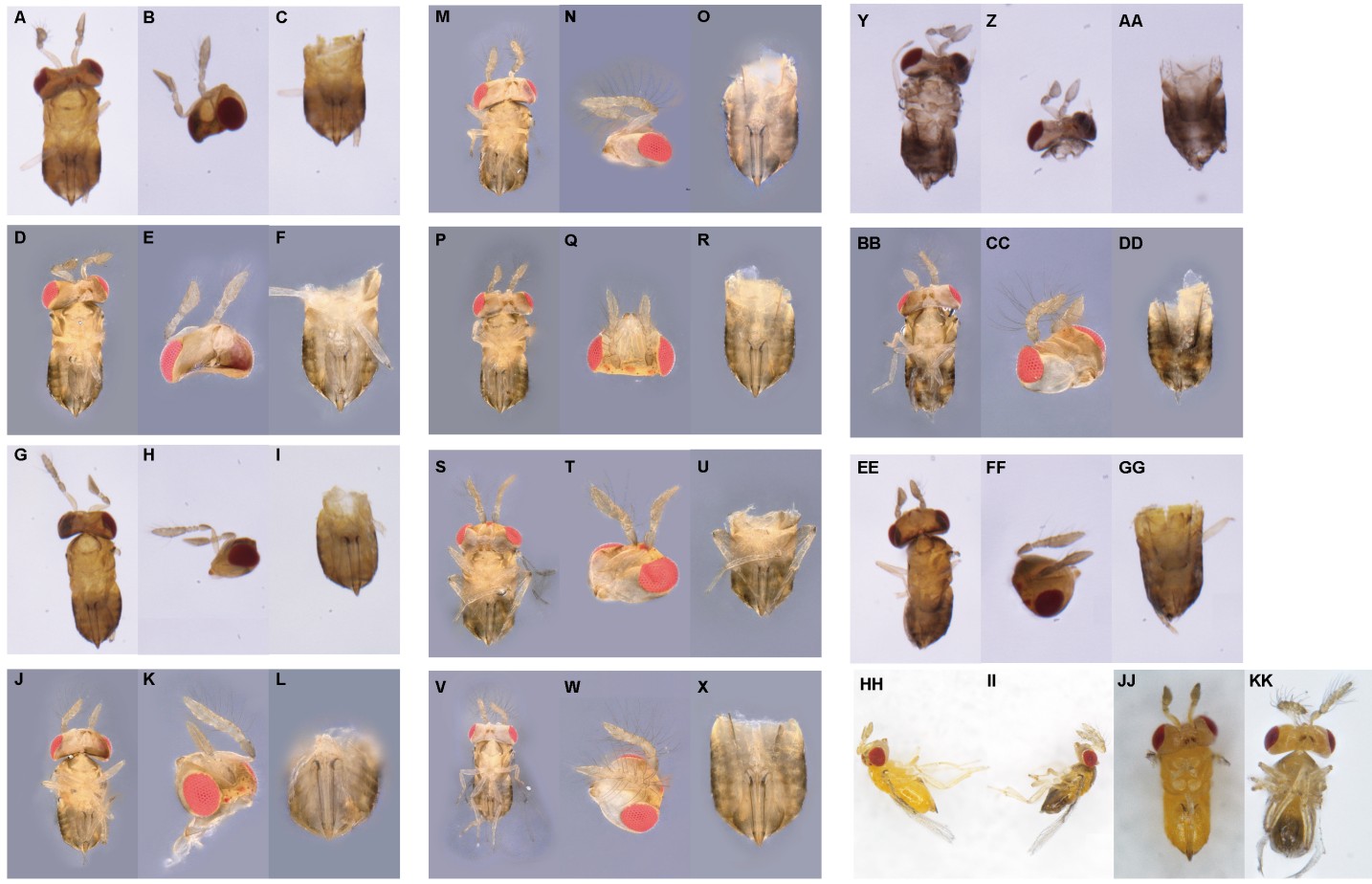

**Figure 1  Description of the external morphology of 11 types of intersexes in *T. preW⁺*.** (1) (A–X) Female intersex, the external genitalia are female and the antennae tend to masculinized. (A, D, G, J, M, P, S, V) represent the ventral (secondary sexual characteristics), (B, E, H, K, N, Q, T, W) represent the antennae (secondary sexual characteristics), and (C, F, I, L, O, R, U, X) represent the external genitalia (primary sexual characteristics), respectively. Type A of intersex: (A–C) (A: the ventral of intersex; B: two female antennae, one with few short setae; C: female genitalia). Type B of intersex: (D–F) (D: the ventral of intersex; E: two female antennae, both with few short setae; F: female genitalia). Type C of intersex: (G–I) (G: the ventral of intersex; H: one male antenna with few short setae, one female antenna; I: female genitalia). Type D of intersex: (J–L) (J: the ventral of intersex; K: one male antenna with few long setae, one female antenna; L: female genitalia). Type E of intersex: (M–O) (M: the ventral of intersex; N: one male antenna with few long setae, one female antenna with few short setae; O: female genitalia). Type F of intersex: (P–R) (P: the ventral of intersex; Q: two male antennae, both with short setae; R: female genitalia). Type G of intersex: (S–U) (S: the ventral of intersex; T: two male antennae, one with few long setae, one with few short setae; U: female genitalia). Type H of intersex: (V–X) (V: the ventral of intersex; W: two male antennae, both with long setae; X: female genitalia). (2) (Y–GG) Male intersex, the external genitalia are male and the antennae tend to feminized. (Y, BB, EE) represent the ventral (secondary sexual characteristics), (Z, CC, FF) represent the antennae (secondary sexual characteristics), and (AA, DD, GG) represent the external genitalia (primary sexual characteristics), respectively. Type I of intersex: (Y–AA) (Y: the ventral of intersex; Z: two female antennae; AA: male genitalia). Type J of intersex: (BB–DD) (BB: the ventral of intersex; CC: one male antenna with long setae, one female antenna; DD: male genitalia). Type K of intersex: (EE–GG) (EE: the ventral of intersex; FF: one male antenna with long setae, one female antenna with few short setae; GG: male genitalia). (3) (HH–KK) As control group, is the normal female in *T. preW* and *T. preW⁺*, normal male in *T. preW*. (HH) Is the lateral of female and (II) is the lateral of male. (JJ and KK) represent ventral of female and male, respectively. The external morphology of female is two female antennae, none of setae, female genitalia. The external morphology of male is two male antennae, both with long setae, male genitalia.

with a span of 30 min. Finally, the samples were treated with pure glycerol twice for 30 min each time. Then, samples were stored in the refrigerator (HYC-940; Haier, Qingdao, China) at 4 °C.

The intersex samples were softened at room temperature. To observe the characteristics of antennae and external genitals, we first removed the wings and feet from the body and then the head and abdomen were cut off by a scalpel under the stereomicroscope lens (SMZ-161; Motic, Xiamen, China). Thereafter, the samples were photographed with the camera (SMZ25; Nikon, Tokyo, Japan).

## Data analysis

The probability of the intersex (binomial distribution) and the level of masculinization of the intersex (Poisson distribution) as influenced by generations and temperature were analyzed by the logistic model and generalized log-linear model, respectively. The multiple comparisons of all treatment pairs were carried by Tukey–Kramer tests. The homogeneity of all models was tested by studentized Breusch–Pagan test and Shapiro test, respectively (*Hall, 1992*; *Koenker, 1981*). All analyses were carried out with R ver. 3.5.1 (*R Core Team, 2017*).

## RESULTS

### General characterization of intersex individuals produced by *T. preW* $^+$

In this study, we found 235 intersexes from 83,699 offspring wasps produced by female of *T. preW* $^+$ infected *Wolbachia* under different temperatures (Supplemental Information S1). Based on external characters, we were able to classify the intersex into 11 morphological categories (A–K) with increasing masculinization (Fig. 1). The external genitalia of most intersexes were female, and only I, J, K intersex types have male external genitalia (Supplemental Information S2). In addition, the intersexes were not found in the population of *T. preW* (Supplemental Information S1).

### The probability of intersex as influenced by temperature

The probability of intersex was significantly influenced by generations ($\chi^2 = 17.16$, $P < 0.001$) or temperature ($\chi^2 = 143.13$, $P < 0.001$), but it was not influenced by the interaction of generations and temperature ($\chi^2 = 0.68$, $P = 0.41$) (Fig. 2). The probability of intersex increased with temperature (Coefficient ± SE = 0.40 ± 0.036, $z = 11.10$, $P < 0.001$). The average probability of intersex in $F_1$ (27 °C: 0.17% ± 0.12% (Mean ± SE); 29 °C: 0.51% ± 0.25%; 31 °C: 1.43% ± 0.40%; 33 °C: 1.41% ± 0.52%) was significantly higher ($z = 4.70$, $P < 0.001$) than it was in $F_2$ (27 °C: 0.07% ± 0.016%; 29 °C: 0.26% ± 0.040%; 31 °C: 0.25% ± 0.029%; 33 °C: 1.06% ± 0.12%).

### The degree of secondary masculinization level of intersex individuals in *T. preW* $^+$

The masculinization level of intersex individuals were significantly influenced by temperature ($\chi^2 = 8.25$, $P < 0.01$), but they were not influenced by generations ($\chi^2 = 0.37$, $P = 0.54$) and the interaction of generations and temperature ($\chi^2 = 0.56$, $P = 0.45$) (Fig. 3). The masculinization level of intersex individuals significant increased with temperature (Coefficient ± SE = 0.039 ± 0.014, $z = 2.85$, $P < 0.01$).

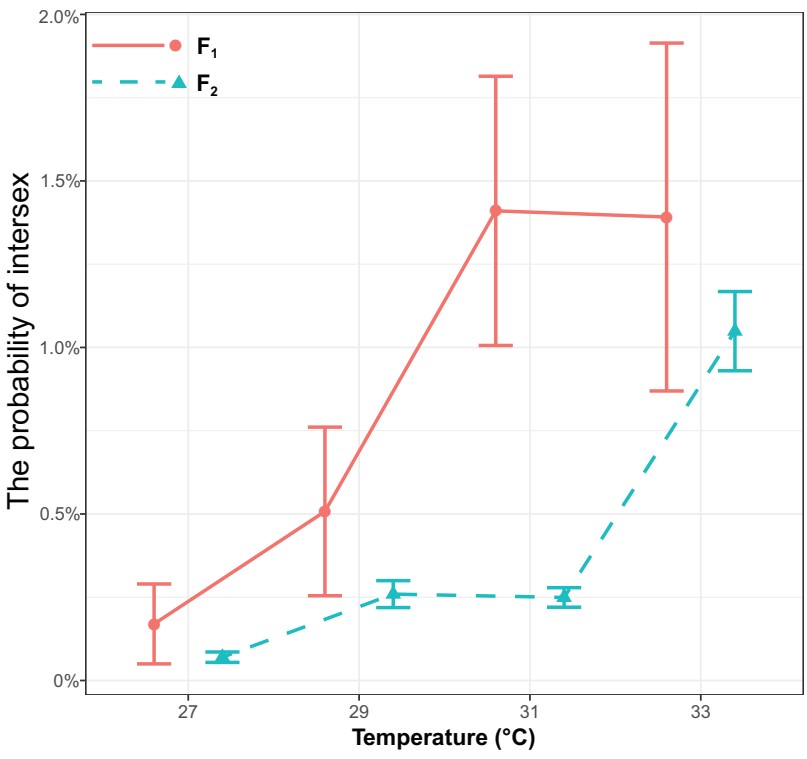

**Figure 2** **The effects of different temperature in the parental generation (27, 29, 31, and 33 °C) on the probability of intersex in the $F_1$ and $F_2$ generation in *T. preW*$^+$.** The blue line indicates the $F_2$ generation and the red line indicates the $F_1$ generation.     

## DISCUSSION

In order to determine the causes of intersex formation in *T. preW*$^+$ and the effects of parental generation temperature of gradual changes in the masculinization of secondary sexual characteristics in intersex offspring, we focused on sexually aberrant intersexes and found that intersex offspring occurs after the *T. preW*$^+$ parental generation experiences high temperature. The parental generation temperature affects the probability of occurrence of intersex offspring and the degree of masculinization in secondary sexual characteristics in intersexes. A large volume of studies has showed that high temperature will decrease *Wolbachia* titers inside *Trichogramma* wasps and produce males and intersexes, thereby changing the original reproduction method and offspring sex ratio in *Trichogramma* wasps (*Pintureau, Chapelle & Delobel, 1999*; *Zchori-Fein, Gottlieb & Coll, 2000*; *Ma & Schwander, 2017*). In addition, it is found that there is no intersex in the offspring in *Wolbachia*-uninfected *T. pretiosum* in this paper. The following conclusions can be obtained: *Wolbachia* is a direct factor that causes the occurrence of intersexes, while high temperature is an indirect factor that determines the external morphology of intersexes.

All intersexes are caused by sex determination and aberrant sexual differentiation. Therefore, a strict classification of intersex types has low significance (*Gusmao & McKinnon, 2009*; *Pereira et al., 2010*). Hence, our study did not compare the external

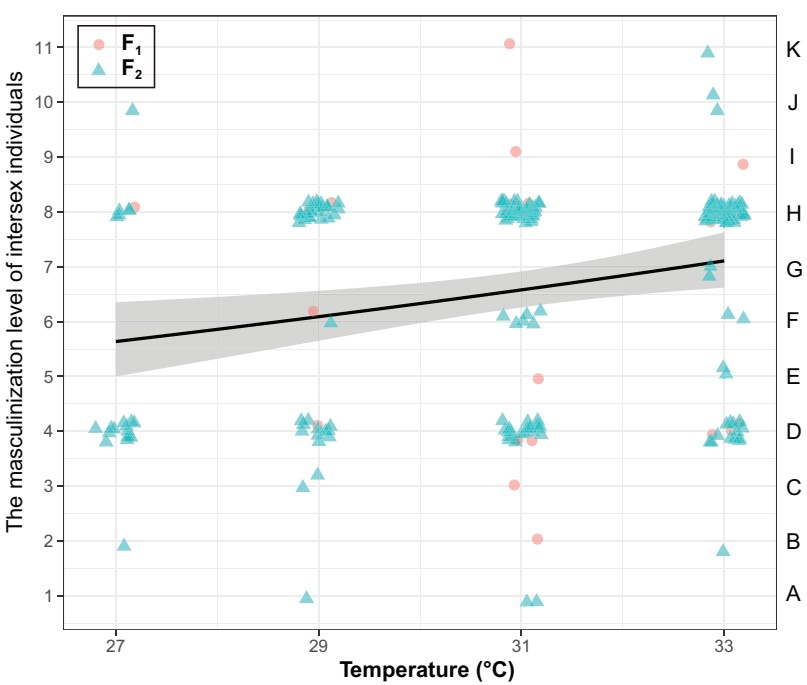

**Figure 3 The degree of masculinization level of intersexes in _T. preW⁺_.** The left ordinate of the graph represents the masculinization level of intersex individuals. The right ordinate of the graph represents the types of intersexes. The shaded area said confidence interval.

morphology of different types of intersexes but examined the relationship between parental generation temperature and different offspring with intersexes through calculating the probability of intersex occurrence. Results showed that _T. preW⁺_ parental generation temperature is positively correlated with the probability of intersex occurrence. The probability of intersex occurrence in the $F_1$ generation is significantly higher than that of the $F_2$ generation. This shows that the effects of high temperature on _T. preW⁺_ intersexes is passed through the parental generation to offspring, and this maternal effect weakens as the number of generations increases. A previous study found that high temperature can affect _Wolbachia_ titers to affect the sex of offspring (_Sakamoto et al., 2008_; _Narita et al., 2007_; _Sugimoto et al., 2015_). If this is true, it is not difficult to imagine that the effect of high temperature on the first generation is stronger, causing lower _Wolbachia_ titers. At this point, the effect of sex determination mechanisms of the host is relatively stronger than feminization induced by _Wolbachia_, and it can causes stronger genetic conflict between _Wolbachia_ and host genes, and the number of intersex individuals increases. In contrast, the weakening of the effects of high temperature on the second generation combined with self-recovery of _Wolbachia_ increases _Wolbachia_ titers. This causes _Wolbachia_ to gradually restore the ability that control host reproductive patterns, and the number of intersex individuals decreases. However, another potential explanation is that intersexes may be induced in some Hymenoptera species with complementary sex determination (CSD) mechanism (_Tulgetske & Stouthamer, 2012_). CSD is a sex-determining mechanism of alleles at sex locus determine the sex of offspring

(*Vorburger, 2014*). However, our unpublished results have showed that the sex determination mechanisms for *Trichogramma* wasps does not conform to CSD but may conform to a form of genome imprinting (QQ Liu, 2018–2019, personal observations). The genome imprinting model has been described in *Nasonia vitripennis* (*Heimpel & De Boer, 2008*). In this model, a maternally active sex-determination gene imprints a sex-determination gene (potentially the same gene) of fertilized eggs. Haploid eggs contain only a maternally derived, and thus imprinted, sex-determination gene and develop into males. Diploid eggs contain a paternally derived nonimprinted copy of the sex determination gene in addition and develop as diploid females. Genomic imprinting is often caused by DNA methylation in organisms, but not yet been found in hymenopteran insects (*Heimpel & De Boer, 2008*). With regards to *T. preW*$^+$ in this study, if we assume that genome imprinting is retained in the offspring by the parental generation and the presence of *Wolbachia* inhibits the effects of genome imprinting, when temperature reduces *Wolbachia* titers, the effects of parental generation genome imprinting will become stronger. This may be the cause of intersex formation.

In this study, the degree of masculinization in primary and secondary sexual characteristics in intersexes was used to identify 11 intersex types. In addition, we found that the frequency of the D (one male and one female antenna, male antennas contain long setae, female antennas do not contain bristles, external genitalia present in females) and H types (two male antennas that possess long setae, female external genitalia present) of intersex was the highest. The external genitalia of most intersex types were female, and only three intersex types have male external genitalia. In *T. preW*$^+$ intersex individuals, most exhibit female primary sexual characteristics, and secondary sexual characteristics exhibit signs of masculinization. *Tulgetske & Stouthamer (2012)* found that the degree of masculinization in secondary sexual characteristics of intersexes is randomly distributed and not associated with high temperature. Another paper has showed that high temperature will cause feminization of secondary sexual characteristics in *Culex stimulans* intersexes with a male genotype (*Anderson & Horsfall, 1965*). In contrast, high temperature can also induce masculinization of secondary sexual characteristics in *T. deion* (*Bowen & Stern, 1966*), *Solenobia triquetrella* and *Carausius morosus* intersexes (*Pereira et al., 2010*). In our study, we found that the degree of masculinization of secondary sexual characteristics in *T. preW*$^+$ intersexes increases as parental generation temperature increases. The differences in results may be due to the differences in sex determination mechanisms in different hosts and the presence of *Wolbachia*. The genetic conflict between host genes and *Wolbachia* may also be the cause of intersex development (*Rigaud & Juchault, 1993*, *1998*; *Stouthamer et al., 2010*).

In summary, high temperature affects *Wolbachia* in the parental generation to affect sex changes in the host offspring, and the maternal effect that causes this phenomenon decreases as the number of generations increases. Many papers on insect intersexes in the order Hymenoptera have focused on external morphology and behavior and less on sex-determining genes. As the frequency of intersex occurrence in *T. preW*$^+$ is low, and *T. preW*$^+$ individuals are small, it is relatively difficult to collect or dissect these intersexes and determine whether all intersex can reproduce. More studies should be carried in

future. Therefore, we only studied the external genitalia of intersexes in this paper. In the future, in-depth research on internal genitalia and the fertility of $T.\ preW^+$ can be carried out to facilitate the understanding of intersex formation mechanisms and a deeper understanding of sex determination and differentiation in the order Hymenoptera.

## CONCLUSIONS

We focused on sexually aberrant intersexes and found that intersex offspring occurs after the $T.\ preW^+$ parental generation experiences high temperature. The following conclusions can be obtained: *Wolbachia* is a direct factor that causes the occurrence of intersexes, while high temperature is an indirect factor that determines the external morphology of intersexes. This change of sex may be associated with the changes in the expression levels of sex-determining genes. The changes in the expression levels of sex-determining genes in $T.\ preW^+$ that are jointly caused by temperature-*Wolbachia* is the root cause for intersex formation. In view of the aforementioned situation, we recommend that more in-depth research on relevant sex-determining gene regulation mechanisms for intersexes be carried out in the future to determine whether changes in the expression levels of sex-determining genes results in intersex formation to elucidate the mechanisms for intersex formation.

## ACKNOWLEDGEMENTS

We thank Su-ying Bao for supplying *Corcyra cephalonica* strains. We thank editors Stephen Johnson and Alex Ford for constructive comments on a previous version of this manuscript.

### Funding

The work was supported by the National Key Research and Development Program of China (2017YFD0201000) and the Natural Science Foundation of Liaoning Province (2015020768). The funders had no role in study design, data collection and analysis, decision to publish, or preparation of the manuscript.

### Grant Disclosures

The following grant information was disclosed by the authors:
National Key Research and Development Program of China: 2017YFD0201000.
Natural Science Foundation of Liaoning Province: 2015020768.

### Competing Interests

The authors declare that they have no competing interests.

### Author Contributions

- Su-fang Ning conceived and designed the experiments, performed the experiments, analyzed the data, contributed reagents/materials/analysis tools, prepared figures and/or tables, authored or reviewed drafts of the paper, approved the final draft.
- Jin-cheng Zhou analyzed the data, approved the final draft.

- Quan-quan Liu approved the final draft.
- Qian Zhao approved the final draft.
- Hui Dong approved the final draft.

## Data Availability

The raw data are available as Supplemental Files.

## Supplemental Information

Supplemental information for this article can be found online at http://dx.doi.org/10.7717/peerj.7567#supplemental-information.

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
