# Peer review of "Gradual, temperature-induced change of secondary sexual characteristics in Trichogramma pretiosum infected with parthenogenesis-inducing Wolbachia"

_PeerJ, doi:10.7717/peerj.7567_

## Round 0.1 · original submission · Minor Revisions

Thank you for your submission to PeerJ. The reviewers have recommended some changes to the manuscript which I'd like you to consider.

Reviewer 1 ·

Basic reporting

The current paper compared the temperature effect on the production of intersex individuals of Trichogramma pretiosum when they are infected with Wolbachia. In this study the authors install reproductive females in different environment (temperature) and measured external parameters to evaluate intersex characteristics.
The results are very concise and clear, and the paper is really focused.
Here are some general comments.
The introduction explains that in the presence of Wolbachia individuals impact the production of females.
L59 and 60: what are the two types of intersex?
L74: the reference Almeida and Stouthamer, 2018 is not appropriated to described Wolbachia impact on insects. Several review or book chapter are more appropriate.

What is the link between the parthenogenesis induced y Wolbachia and the observed intersex? The connection in the introduction is difficult.
L79: the word feminization here is not appropriated. Diploid individuals are necessary females, right?
L81-82: the feminization here refers to individuals that are genetically males with the physiology of female, such process is very different that the one observed in the current study model.
Could the authors clarify the definition of intersex in their species? Morphology? Physiology impact? Can they reproduce? Life history traits?
In the introduction, the authors didn’t introduce the concept of masculinization, but this part is really developed in the result section and in the discussion.

Experimental design

Materials and methods.
L113: 20 generations: how long is one generation? Why did the authors wait for so long? Was the raising condition managed in a specific way? Was the reproduction controlled?
When the individuals were killed, how old were they? How were they scarified?
Could the authors explain why the preparation L134-137 was necessary as they observed external aspect of the individuals?
L139: remove “for”.

Validity of the findings

Results:
Table 1: the authors could also add the “letter” for each line (A to K) it will be easier to compare the different categories.
I suggest to add pictures of antennae and genitalia of regular female and regular male as a comparison. What is the main difference between male and female antennae?
Moreover, the pictures in the table are too small, it’s almost impossible to see the sample.
L154: here for the first time the use of the word: masculinization is surprising as since the beginning we read about feminization. Same comment l165.
The authors should also include the results with asymbiotic individuals and it’s a control. Including in the figure and or legends. It’s mentioned only L181 of the discussion.
L204: what is CSD?
Why the authors switch in the discussion to masculinization, I don’t understand, the underline process.
Figure 1. The figure showed probability. Is it a prediction or based on your real data?

Reviewer 2 ·

Basic reporting

I would suggest corrections on the English language. For example, I don't like the word "proven" that is often used throughout the text, because it is often a too strong statement (replace by "this study showed", or "this study suggests"...).
Else, the article structure is fine to me.
In the discussion, I have one major concern. Authors claimed that "Wolbachia is
a direct factor that causes the occurrence of intersexes, while high temperature is an indirect factor that determines the external morphology of intersexes". However, it is hard to be so affirmative. Indeed, the effect of temperature is supposed to be through an effect on Wolbachia (partial destruction or inhibition of their effects). To be affirmative, one should expose the W+ lines to a temperature that have no effect on Wolbachia, e.g. 25°C. This would have allowed to present the "basal" proportion of intersexes (i.e. proportion of intersexes with Wolbachia in absence of any effect of high temperatures). Here this control series is lacking.
I would like, also, that authors note if intersexes are sterile or not. This is important, because if they are sterile, the impact of the interaction Wolbachia*Temperature, by increasing the proportion of intersexes, would induce a cost for the Trichogramma.

Experimental design

no comment

Validity of the findings

Figure 1 is showing that the proportion of intersexes reach a maximum of 1.5%. However, in the text (L. 162-163), the authors noted that the "The probability of intersex in F1 is 0.66±0.13 %... and in F2 0.24±0.019 %". There is therefore a huge difference between the text and the figure. Please correct this.

---

## Round 0.2 · Minor Revisions

Dear Authors, thank you for submitting your revised manuscript. The reviewers have suggested a few more changes to the sentence structures and have a few more questions to clarification and aid the reader.

Reviewer 1 ·

Basic reporting

The authors addressed most of the comments and improved it but there is still some point that could be clarified.

What are the consequences of Wolbachia in the host species? Of course beside parthenogenesis? Do you have any information regarding other impacts?
Do you have information regarding Wolbachia prevalence in natural population?
Do the authors know if all intersex can reproduce?
In the answer you explain the complexity of the sex determinism of the species which is very important and could help the reader to better understand your study model and the impact of Wolbachia. This information could also contribute to better understand the discussion L 223-225.
L82: “known” should be corrected.





Discussion.
L198 -199: could we considered that without Wolbachia the sex ration is 50/50 for males and females in the study species? Or is it almost females?
L215 “… is relatively stronger,..”. Could the author precise stronger than what?
L217: the verb of the sentence is missing
L219: the authors should precise what they mean by dominance.

Experimental design

Materials and methods.
Did the authors check for Wolbachia infection in some individuals? Are they sure that 100% of the females were Wolbachia infected?

L115: “genenrations” should be generations and “collectiong” should also be changed.
L153: authors did some dissection but never said which dissection? Dissection to check what, which tissue? And do they have any picture?
L155 the sentence should be rephrased

Validity of the findings

Results:
The authors found 235 Intersex. Could the authors precise out of how many “normal” individuals?
The probability are very low and the value (%) are difficult to understand if the authors do not provide more information.
These informations are provided in the supplementary materials, I suggest that the authors refer to the supplementary data in the text.
Moreover, in the supplementary file, the intersex are describe with number but in the manuscript it’s letters. Could the authors add the letter in the supplementary file as well.

Reviewer 2 ·

Basic reporting

The authors responded satisfactorily to my comments.

Experimental design

N.A.

Validity of the findings

The authors responded satisfactorily to my comments.

Additional comments

I noted few language oddities:
- L. 77-78: I do not really understand what you want to say. is it : "According to our knowledge of Trichogramma, intersexuals are only present in parthenogenetic species."
- L. 113: change "genenrations " for "generations", and "collectiong " for "collection"
- L. 117 should "contains " be "containing" ?
- L. 143-144. There is something wrong with the sentence "In order to obtain the pictures of all masculinization levels of intersex and make the size of the sample remains unchanged for a long time." As far as I understood, it should be an introductory part of the following sentence (?). So the dot should be replaced by a comma.
L. 246-247 : the sentence "The genetic conflict between host genes and Wolbachia-induced parthenogenesis is the cause of intersex can induce intersex to occur" is weird. It should be something like : "The genetic conflict between host genes and Wolbachia may also be the cause of intersex development".

---

## Round 0.3 · accepted · Accept

Dear Authors, thank you for sending through your revised manuscript. I’m delighted to accept your interesting manuscript and thank you for submitting to PeerJ.